# The antenatal psychological experiences of women during two phases of the COVID-19 pandemic: A recurrent, cross-sectional, thematic analysis

**Leanne Jackson[1], Siân M. Davies[2,3], Anastasija Podkujko[1], Monic Gaspar[1], Leonardo L. D. De Pascalis[1], Joanne A. Harrold[1], Victoria Fallon[1], Laura K. Soulsby[1‡], Sergio A. Silverio[3‡]***

1 Department of Psychology, Institute of Population Health, University of Liverpool, Liverpool, Merseyside, United Kingdom, 2 School of Psychology, Faculty of Health, Liverpool John Moores University, Liverpool, Merseyside, United Kingdom, 3 Department of Women & Children's Health, School of Life Course & Population Sciences, King's College London, Southwark, London, United Kingdom

‡ LKS and SAS are joint senior authors on this work.
* Sergio.Silverio@kcl.ac.uk

**Data Availability Statement:** The datasets used and/or analysed during the current study are part of a common dataset from The PRaM Study. The

## Abstract

Initial COVID-19-related social distancing restrictions, imposed in the UK in March 2020, and the subsequent lifting of restrictions in May 2020 caused antenatal disruption and stress which exceeded expected vulnerabilities associated with this lifecourse transition. The current study aimed to explore the antenatal psychological experiences of women during different phases of pandemic-related lockdown restrictions in the UK. Semi-structured interviews were held with 24 women about their antenatal experiences: twelve were interviewed after the initial lockdown restrictions (Timepoint 1; T1), and a separate twelve women were interviewed after the subsequent lifting of those restrictions (Timepoint 2; T2). Interviews were transcribed and a recurrent, cross-sectional thematic analysis was conducted. Two themes were identified for each timepoint, and each theme contained sub-themes. T1 themes were: 'A Mindful Pregnancy' and 'It's a Grieving Process', and T2 themes were: 'Coping with Lockdown Restrictions' and 'Robbed of Our Pregnancy'. COVID-19 related social distancing restrictions had an adverse effect on women's mental health during the antenatal period. Feeling trapped, anxious, and abandoned were common at both timepoints. Actively encouraging conversations about mental wellbeing during routine care and adopting a prevention opposed to cure attitude toward implementing additional support provisions may serve to improve antenatal psychological wellbeing during health crises.

## 1. Introduction

### 1.1 Prenatal vulnerabilities

Whilst pregnancy is usually a source for positive emotionality, the risk associated with early pregnancy complications, losses, and changing identity and family dynamics, can provide

datasets are not publicly available due to the sensitive nature of the interviews however they are available upon reasonable request from the team at: PSAS@liverpool.ac.uk.

**Funding:** The author(s) received no specific funding for this work.

significant stress to women's experiences. Pregnant women in particular, are at a heightened risk of experiencing psychological distress [1], and complex psychological and physiological changes which occur during pregnancy can impart additional mental health vulnerabilities [2]. Stressful life events (experienced pre- or post- conception) also exacerbate risk of experiencing distress, such as Post Traumatic Stress Disorder (PTSD) [3]. Healthy lifestyle behaviours e.g., taking regular physical exercise and having an optimistic worldview are related to more positive experiences of pregnancy [4]. Social support can be an important buffer against distress: feeling well supported by non-judgemental peers e.g., one's romantic partner, friends, and/or the wider family unit, is an essential determinant of psychological wellbeing during pregnancy [5]. Whereas feeling ill-supported, judged, and alone are associated with adverse mental health outcomes [6]. The interplay between biological predisposition, life experience, and psychological vulnerabilities are complex and effect severity of antenatal mood alterations [7].

## 1.2 A global health crisis

The novel SARS-CoV-2 outbreak ('COVID-19') [8] was first discovered in Wuhan, Hubei Province, China [9]. The WHO first noted reports of viral pneumonia in China on the last day of 2019, and within a month the virus had rapidly spread causing a WHO declaration of Public Health Emergency of International Concern status for the virus [10]. On 11 March 2020, The WHO declared pandemic status for the COVID-19 outbreak [11], by which time there were more than 118,000 cases and almost 4,300 deaths in 23 countries [12]. Following this declaration, worldwide enforcement of preventative measures, such as: national lockdowns; self-isolation/quarantine; and social distancing restrictions were put in place, with an aim to contain the spread of the virus [13]. Due to the highly infectious nature of the virus, combined with its unpredictable impact on patients, the UK Government implemented a *'national lockdown'* of social distancing restrictions in the interest of protecting public health and reducing NHS strain, which commenced on 23 March 2020 [14, 15]. Whilst COVID-19 infection is experienced with mild symptomatology for most of the public (1% mortality rate) [16], pregnant women were initially determined to be a vulnerable category and were advised to *'shield'* [17]. In the UK, shielding meant not leaving one's home under any circumstances unless seeking urgent medical care, and whilst pregnant women were eventually declassified from this level of vulnerability, it has been found that third trimester COVID-19 infection causes worsened maternal symptomatology [18]. Subsequent evidence countered the precautionary measures given for pregnant women to shield–finding limited evidence for vertical transmission between mother and infant [19], and mild presentation of symptoms in a large proportion of infected infants [20]. Despite which, separation of infected mothers and infants was still routinely practiced in some NHS settings at this time [21]. Lockdown restrictions have had an adverse effect on the psychological health of the public, globally [22]. In the UK, social distancing restrictions have elevated prevalence of depression and anxiety, in general and perinatal populations [23, 24]. Evidence has also suggested that persons advised to *'shield'* were more susceptible to the psychological impacts of the pandemic due to increased restrictions placed on personal freedoms [25].

## 1.3 COVID-19 and prenatal vulnerabilities

Pregnant women experienced a unique set of challenges during the pandemic, which increased risk of experiencing mental distress [26]. Uncertain risks of illness transmission to one's infant or family increased stress and anxiety [27]. Antenatal and services for children aged 0–5 were deprioritised in support of the pandemic response [28] and outpatient clinics were temporarily closed to prioritize the treatment of COVID-19 patients, which left many women without access to time-sensitive maternal health care [29, 30]. Restrictions placed on number of

visitors, length of visiting hours, and opening times for pre-existing services were also implemented to minimize unnecessary exposure to COVID-19 [31, 32]. In support of these efforts, many healthcare services moved online, though this was ineffective for building the necessary rapport to disclose mental health concerns for pregnant women [33–36]. Access to community healthcare, health visiting, and perinatal mental health services were disrupted and compounded with dissipated access to formal childcare services, ability to work away from the home, informal support (e.g., grandparents), and ability to engage in recreational activities outside the home [37–39]. Disrupted access to formal and informal support services have been related to poorer treatment rates for non-COVID related health complaints; increased prevalence of stress, anxiety, depression, and loneliness among pregnant women; and heightened risk of preterm birth, developmental, and psychiatric disorders among infants [35, 40–42].

### 1.4 The current study

There is limited qualitative literature exploring the psychological experiences of antenatal women during the early phases of the COVID-19 pandemic. A large, national on-line survey study conducted by the PRegnancy and Motherhood [PRaM] research team found elevated levels of postnatal depression and anxiety during initial stages of the COVID-19 pandemic i.e., between March 2020 –July 2020 [23]. Furthermore, in recurrent, cross-sectional thematic analyses to elucidate these findings, attributing restricted access to formal and informal sources of support, untailored social distancing guidance, and consequential feelings of isolation, abandonment, and burnout to be contributing factors to observed elevated levels of distress [34, 38]. The current study uses recurrent, cross-sectional thematic analysis to identify specific factors which contributed towards antenatal mental health and psychological wellbeing during the early phases of the pandemic, in an attempt to identify avenues for improved mental health support for expectant mothers now, and in future health crises.

## 2. Materials and methods

Ethical approvals were sought and granted from the University of Liverpool Research Ethics Committee 7 April 2020 [ref:- IPHS/7630]. Verbal, audio recorded consent was taken before the commencement of each interview to ensure the participant was still happy to participate in the current study. Participants were made aware of their right to withdraw and were fully debriefed after interviewing.

### 2.1 Participants

Eligible pregnant women who completed a related national online survey study [23] were redirected to a secondary Qualtrics survey. The secondary survey allowed the woman to provide her contact details if interested in talking about their pregnancy experience during the COVID-19 pandemic in greater depth, as part of an audio-recorded interview study. Expressions of interest were over-subscribed (N = 72). As such, participants were selected for initial contact using an online random number generator. A member of the research team [SMD, MG, AP] contacted the mother approximately 48 hours after receiving an expression of interest, to arrange a time and date for interviewing, and to send the mother an electronic consent form to sign.

Eligibility criteria were consistent with the PRaM national survey [23]: maternal age above 18, third trimester of pregnancy (>28 weeks; to ensure they had some prior experience of maternity care before pandemic restrictions were brought in), English-speaking, and UK-residing. In total 24 participants were recruited, twelve women at Timepoint 1 [T1] and a separate twelve women at Timepoint 2 [T2]. Although we were initially over-subscribed,

participation exhaustion at time of data collection [43] may have rendered recruitment difficult, with many who were initially interested subsequently not wishing to take part due to the chronicity of the pandemic circumstances. Due to the nature of childbirth, some women who were booked for interview gave birth before their allotted slot. To remedy issues regarding participant attrition, these postpartum women who had given birth in the previous four weeks were subsequently included in antenatal recruitment, but asked to reflect solely on their antenatal experiences. In total, eight antenatal participants and four postpartum participants were recruited for the current study, but all women were asked to reflect solely on their antenatal experiences. Please see Table 1 for a summary of T1 and T2 participant characteristics.

## 2.2 Data collection

Recruitment for T1 interviews commenced approximately thirty days after the introduction of initial social distancing restrictions (23 March 2020) and recruitment for T2 interviews commenced approximately thirty days after the first initial easing of social distancing restrictions (11 May 2020). The idea behind these thirty-day 'washout' periods was to prevent any contamination of changes in practices and care delivered at the point of policy change and healthcare delivery which may have marred individual experiences. Individual semi-structured interviews were conducted via telephone or video-calling [SMD, MG, AP]. Interviews lasted between 68–114 minutes (Mean = 93minutes). Interview schedules were created with members of the research team who had expertise in perinatal mental health [VF, SAS, JAH, LDP]. Interview questions had a chronological structure so to conduct an in-depth exploration of maternal experiences through different phases of social distancing restrictions. Data collection followed best practice guidance [44]. Audio recordings were transcribed and analysed in NVivo 12 [MG, AP, LJ, SAS]. All respondents were reimbursed £10 and debriefed.

## 2.3 Analysis

Transcripts were analysed using recurrent, cross-sectional, thematic analysis [47]. Firstly, thematic analysis was used to determine thematic structures at each timepoint, independently [48]. This involved six stages: familiarisation with transcripts, generation of initial codes, identification, review, and defining themes, and report writing [48]. Next, comparisons were made between thematic structures, considering pre-existing perinatal literature [47]. Data saturation was achieved after analysing seven (T1) and eight (T2) transcripts, respectively [49]. Despite data saturation having been assumed earlier, recruitment, interviewing, and analysis continued until our target sample of 24 women had been reached [50]. All authors were involved in refining and identifying themes, following an inductive and consultative approach [51, 52]. Themes are outlined and discussed in detail with supporting illustrative quotations, accompanied by their associated timepoint (i.e., T1 for Timepoint 1, and T2 for Timepoint 2). Illustrative quotations have been pseudo-anonymised throughout for the purpose of protecting participant confidentiality.

## 3. Results

We present results from two timepoints during the pandemic. Timepoint 1 [T1] data reflects the antenatal psychological experiences of women during the initial lockdown across the UK and is presented as a full analysis first. Timepoint 2 [T2] data reflects the antenatal psychological experiences of women during the initial lockdown easing in the UK, and is presented second, as a full analysis in its own independent right. See Table 2 for a breakdown of the themes and sub-themes derived from the two datasets. A comparison between T1 and T2 data is then presented in the discussion.

**Table 1. Demographic characteristics.** (Occupation and Education categories taken from UK Government [45, 46].

| Timepoint | Participant Pseudonym | Age (Years) | How Many Weeks Pregnant at Time of Interview? | Occupation | Highest qualification | First child? |
|---|---|---|---|---|---|---|
| T1: Initial Lockdown Restrictions | Ethel | 24 | 39 | Caring, leisure and other service occupations | Secondary School Education (incl. ALevels and BTECs) | No |
| | Madeline | 26 | 31 | Sales and customer service occupations | Secondary School Education (incl. ALevels and BTECs) | No |
| | Georgina | 28 | 30 | Skilled trades occupations | Secondary School Education (incl. ALevels and BTECs) | Yes |
| | Ugne | 29 | 35 | Associate professional occupations | Master's Degree | Yes |
| | Lindsay | 29 | 38 | Managers, directors, and senior officials | Doctoral degree | No |
| | Danni | 32 | 31 | Professional occupations | Bachelor's Degree (incl. Medical Degree) | Yes |
| | Claire | 33 | 33 | Professional occupations | Bachelor's Degree (incl. Medical Degree) | No |
| | Kavitha | 33 | 33 | Managers, directors, and senior officials | Bachelor's Degree (incl. Medical Degree) | Yes |
| | Olivia | 34 | 39 | Managers, directors, and senior officials | Bachelor's Degree (incl. Medical Degree) | Yes |
| | Emily | 35 | 33 | Professional occupations | Bachelor's Degree (incl. Medical Degree) | No |
| | Ceinwen | 37 | 35 | Administrative and secretarial occupations | Master's Degree | No |
| | Miriam | 39 | 32 | Professional occupations | Doctoral degree | No |
| | Bashita | 26 | 37 | Associate professional occupations | Master's Degree | No |
| T2: Lifting of Lockdown Restrictions | Xenia | 30 | 37.5 | Professional occupations | Master's Degree | Yes |
| | Joely | 30 | Postnatal [a] | Professional occupations | Bachelor's Degree (incl. Medical Degree) | No |
| | Lilian | 31 | 39 | Professional occupations | Doctoral degree | Yes |
| | Aliyah | 32 | 38 | Caring, leisure and other service occupations | Doctoral degree | No |
| | Francine | 32 | Postnatal [a] | Associate professional occupations | Bachelor's Degree (incl. Medical Degree) | Yes |
| | Francesca | 33 | 35 | Professional occupations | Bachelor's Degree (incl. Medical Degree) | Yes |
| | Emma | 33 | 37 | Professional occupations | Bachelor's Degree (incl. Medical Degree) | No |
| | Selina | 33 | Postnatal [a] | Professional occupations | Doctoral degree | No |
| | Chloe | 34 | 32 | Administrative and secretarial occupations | Bachelor's Degree (incl. Medical Degree) | No |
| | Daisy | 35 | Postnatal [a] | Managers, directors, and senior officials | Doctoral degree | No |
| | Zanthia | 41 | 33 | Associate professional occupations | Bachelor's Degree (incl. Medical Degree) | Yes |

[a] **N/B.** Whilst these women were postnatal at the time of interview, they were asked to reflect on their antenatal experiences only.

## 3.1 T1 results

The T1 thematic analysis generated two main themes, each with two sub-themes. '*A mindful pregnancy*' was the first identified major theme: capturing how social distancing restrictions had enabled pregnant women to engage in their pregnancy journey in a more present and

**Table 2. Themes and sub-themes at both timepoints.**

| Timepoint 1 (T1)–Initial Lockdown Restrictions in the UK (Commenced 23 March 2020) | | Timepoint 2 (T2)–Initial Easing of Lockdown Restrictions in the UK (Commenced 11 May 2020) | |
|---|---|---|---|
| **THEME** | *SUB-THEME* | **THEME** | *SUB-THEME* |
| **A Mindful Pregnancy** | *Listening to Your Physical and Emotional Needs* | **Coping with Lockdown Restrictions** | *A Slower Pace of Life* |
| | *Controlling What You Can and Disengaging from What You Can't* | | *Being Kind and Keeping Structure* |
| **It's a Grieving Process** | *Lonely and Anxious* | **Robbed of our Pregnancy** | *Unmet Expectations and Feeling Guilty* |
| | *Lost Freedoms and Unmet Expectations* | | *The Voice of Others Feels Loud* |
| | | | *Left in the Dark* |

purposeful fashion. *'It's a grieving process'* was the second major theme: elevated feelings of imprisonment and anxiety were expressed in relation to restrictions enforced on personal freedoms. Many T1 respondents grieved their lost pregnancy experience and were saddened by the overbearing impacts of the pandemic in undermining their antenatal journeys. Indeed, antenatal accounts were consistently felt to have been undermined by pre-occupations of friends and family with the COVID-19 pandemic. The T1 thematic structure was as follows:

1. A Mindful Pregnancy

1.1. Listening to Your Physical and Emotional Needs

1.2. Controlling What You Can and Disengaging from What You Can't

2. It's a Grieving Process

2.1. Lonely and Anxious

2.2. Lost Freedoms and Unmet Expectations

## Theme 1: A mindful pregnancy

A slowed pace of life, due to restrictions on personal freedoms, granted mothers improved bonding with their foetus and the ability to listen with kindness to the needs of their body. Women expressed also making mindful decisions about exposure to media portrayals of the pandemic, and made pro-active efforts to maintain their emotional wellbeing, exert control where possible, and to exercise acceptance for the unprecedented nature of the pandemic.

**Sub-theme 1:** Listening to your physical and emotional needs

The dissipation of pre-COVID responsibilities and busy schedules gave respondents the opportunity to bond more deeply with their unborn baby:

"I feel a bit closer to the baby than I maybe would. Well, it's hard to know isn't it, but I have a lot of time by myself and I talk to the bump." [Danni, T1]

This slowed pace of life was welcome to most respondents, who valued having the unique opportunity to rest and recuperate:

"Not many people at seven, eight months pregnant get this time to be at home and listen to their body and be able to relax when they can. If I'm tired, I can go for a lie down." [Georgina, T1]

It appears that trusting one's body and ability to birth, drawn from previous experience, allowed respondents to remain autonomous and self-assured:

"I've attended the [hospital] appointments, but they've been less important to me because I've been relying a little bit on the fact that I know what the pregnancy feels like, I know what my body's doing, it feels very similar to last time." [Lindsay, T1]

For some, even more challenging aspects of early pregnancy e.g., morning sickness, were welcome assurances to women that their bodies were healthy:

"And the morning sickness actually gave me a bit of confidence because you feel like things are happening as they should and you are experiencing the symptoms of pregnancy." [Olivia, T1]

Being active and taking care of one's physical as well as emotional needs were salient in the accounts of numerous respondents:

"My friends, family, very much so. Exercise I've found really good. I started kickboxing quite a while ago and then got into that quite a lot more when I was feeling down, back around 2016/17 time, and that I found really useful, and then the exercise itself and all of the feelings that gives you." [Kavitha, T1]

Acknowledgment was also made to the more difficult emotional periods:

"I have just been here doing a bit of online teaching and baking and enjoying myself. Then other days I am angry for everything I have missed." [Ugne, T1]

**Sub-theme 2:** Controlling what you can and disengaging from what you can't
Respondents found active coping in immersing themselves in preparations for baby's arrival:

"Pregnancy is a very practical thing, so it was a matter of just ticking all the boxes, looking after myself as best I can, trying not to overdo things or overstress myself" [Lindsay, T1]

Though for others, having little variation in one's routine was perceived to be monotonous:

"My days look like Groundhog Day, mostly revolving around feeding people." [Miriam, T1]

Distracting oneself by setting and achieving manageable goals helped to maintain a sense of accomplishment and wellbeing:

". . .feeling like I have achieved something every day. I have a daily list of things that I try and do because I know it helps my mental health. Doing at least a 3k walk every day. If I don't manage 3k it doesn't matter, but I go out of the house, walk around the block and come back. Do part of a yoga class online, whether it is a five-minute stretch all the way through to a 50-minute session. Engage with something around that. Not trying to set too high expectations for myself, but also be like, you have achieved this" [Ugne, T1]

Active efforts were taken to avoid and distract from uncontrollable negative influences which were not self-serving:

"I usually just distract myself. Play with my phone or watch TV. Talk to somebody if I had a friend who I felt I was able to talk to at the time." [Claire, T1]

Dissociating oneself from the negativity of daily COVID-19 reporting's was protective of emotional wellbeing:

"Try and turn the news off, get little bits here and there if you want to but don't become totally consumed with it because it can be really overwhelming." [Georgina, T1]

**Theme 2:** It's a grieving process
Inability to interact with other adults as a consequence of employment arrangements exacerbated feelings of entrapment. Increased feelings of anxiety were attributed to uncertainties relating to the duration of implemented restrictions, though many respondents were invalidating of their experiences and often posed difficulty distinguishing between normal pregnancy-related anxieties and COVID-19 related uncertainties. One's inability to engage in pregnancy-related rituals e.g., baby showers, with loved ones and perceived lack of recognition from loved ones regarding one's pregnancy was a severe source of loss. Insufficient access to quality perinatal mental healthcare services, too, was a deepened source of frustration and perceived abandonment expressed by pregnant respondents.

**Sub-theme 1:** Lonely and anxious
Social distancing restrictions triggered feelings of entrapment and helplessness:

"I hadn't seen anybody for weeks and weeks and then lockdown happened and I felt quite alone then, and quite isolated and a little bit trapped." (Ethel, T1)

Heightened anxiety was relayed in connection with feeling isolated from one's loved ones and restricted in one's personal freedoms:

"I have other days where I'm struggling and I'm missing my family and my friends and doing all the things that I used to do and just feeling really anxious about everything. . ." (Georgina, T1)

For respondents who were not in paid employment, limitations placed on ability to interact with others left respondents feeling without purpose:

"I definitely find that I'm drifting more, I think, especially if you don't have work or anything like that. I think that's maybe how a lot of people feel, they feel as though they're drifting." (Ceinwen, T1)

Some respondents took precautionary measures to prevent COVID-19 infection, such as avoiding face-to-face contact with others. However, this was a source of anxiety when respondents considered the uncertain duration of social distancing restrictions:

"So, in my mind I'm going to be fairly isolated, not in official isolation terms, but I think I'm going to be keeping myself to myself right through the year, possibly until we get a

vaccine [laughs]. Of course, that's a big commitment, because we actually have no idea when that could be." (Lindsay, T1)

Several respondents were conscious of their attempts to exert control in attempt to remedy pressing global uncertainties of the pandemic:

"I have definitely been searching for more control over things because I feel so out of control of everything to do with this [pandemic] situation." (Ugne, T1)

Despite extraordinary stressors imposed by social distancing restrictions, respondents struggled to distinguish between '*normal*' pregnancy-related anxieties and emotional distress felt due to social distancing restrictions:

"My anxiety has got worse. . . I've referred myself for counselling. Obviously, I'll have to access that over the phone now but. . . I'm not sure whether that's just because my due date is getting closer and I'm more anxious about the birth or whether that is the COVID situation or a bit of both." (Claire, T1)

**Sub-theme 2:** Lost freedoms and unmet expectations
Grief was expressed by respondents who were experiencing difficulty in processing their inability to share their pregnancy with loved ones:

"I feel like I have gone through a grieving process with it. I am grieving the fact that I am not going to get a baby shower. I had a virtual one, but my friends aren't going to get to feel my baby kick." (Emily, T1)

Other disappointments included not having had the opportunity to engage in pregnancy-related rituals and celebrations with others, which would have been a possibility pre-pandemic:

"I have missed out a little bit on–this sounds a bit vain–the attention that you normally get when you're pregnant. When you go out into public places and people are chatting to you and asking questions and that sort of thing." (Claire, T1)

Other women expressed deep sadness and disappointment that their loved ones had not performed acts of kindness in celebration of their pregnancy:

". . .some of the women in my groups have had surprise social distancing baby showers where various different people have dropped things off and waved at them from the car window all excitedly. Dropping off gifts and things like that. Which sounds really lovely. Whereas nobody has done anything like that for me." (Claire, T1)

Unmet pregnancy expectations led women to feel sorrowful and apprehensive about future pregnancies:

"I hope if I ever were to be pregnant again that it would be a less lonely experience. It's difficult because it's not something anyone could have planned. Had I known it would be like this in the early days, maybe I would have made a different decision for now. I don't know. I just hope it wouldn't be like it again." (Kavitha, T1)

Inadequate access to free mental health support was also a deep source of frustration for T1 respondents:

"...access to any mental health services is nigh on impossible really, unless it's something you can source and fund yourself. I did some hypnotherapy that I found myself which was useful, this was a few years ago. She said she was a hypnotherapist but in my mind it was more mindfulness, relaxing, things to help you relax your mind at bedtime." (Kavitha, T1)

## 3.2 T2 results

The T2 thematic analysis generated two main themes, with two and three sub-themes, respectively. As with T1, T2 respondents had found solace in the extra free time which they had been granted to interact more purposefully with their loved ones in the absence of pre-pandemic busy personal and work schedules. T2 respondents were also aggrieved by their overshadowed pregnancy: women were disappointed by their inability to connect with others positively in sharing their personal journey and were frequently reminded of their negative emotional experiences, which exacerbated feelings of distress. Overstated, risk-sensitive media portrayals of the pandemic were anxiety-producing for T2 women and desires were expressed for personalised perinatal mental healthcare services to support emotional wellbeing during times of unprecedented national strain. The T2 thematic structure was as follows:

1. Coping with Lockdown Restrictions

1.1. A Slower Pace of Life

1.2. Being Kind and Keeping Structure

2. Robbed of our Pregnancy

2.1. Unmet Expectations and Feeling Guilty

2.2. The Voice of Others Feels Loud

2.3. Left in the Dark

**Theme 1**: Coping with lockdown restrictions

The furlough scheme was a welcome support for T2 respondents who were enabled to spend more quality time with their romantic partner, and deepened bonding with one's foetus. T2 respondents found self-care activities, especially outdoors, essential for maintaining emotional wellbeing during national lockdown restrictions. Importantly, more available free time facilitated healthier lifestyles to be attained for T2 women, and a more mindful approach to be taken to one's day to day activities. More broadly, social distancing restrictions gave participants a stronger sense of global responsibility and connectivity. Respondents pro-actively engaged in home improvement activities, preparations for baby's arrival, and self-development activities in effort to maintain emotional wellbeing and to exert control amidst the uncertainties of the novel COVID-19 disease. Rebuilding a structure to one's days proved beneficial for maintaining antenatal psychological wellbeing and appeared protective from perceived feelings of loneliness and entrapment due to pandemic related uncertainties.

**Sub-theme 1**: A slower pace of life

For T2 respondents, the furlough scheme allowed new mothers to appreciate a more present and stress-free existence:

"The pros [of lockdown] are I haven't been working like. . . I have a stressful job and I haven't had to deal with the stresses of my job as much, and I've been at home, I've had lots of quality time with my husband. . ." (Xenia, T2)

A slower pace of life also allowed deeper connection to be achieved with one's foetus:

"I've got more time to bond with my bump because I'm in the house a lot more, so that side of it is nice." (Zanthia, T2)

Physically, restrictions also gave pregnant women new opportunities to immerse in healthier lifestyles:

". . .prior to COVID and lockdown actually my job was very stressful and commute and the hours on the job I was thinking were probably not particularly good for me or for the baby, and the things have actually shaken out I'm probably physically healthier now than I have been for years working as a doctor actually because I've been able to look after myself a bit better, so that's finding a positive that I've been able to take from it." (Lilian, T2)

Lockdown restrictions gave participants a newfound appreciation for a simpler and more sustainable lifestyle which bettered emotional wellbeing:

"I think I need to take the simple things in life not for granted, appreciate them more, enjoy the outdoors more, be less disposable as I said and reuse things more. And think about what we spend our money on, and I definitely would love to go on a family holiday and really save up for that and look forward to it and not take it for granted." (Emma, T2)

Meanwhile, restrictions also provided respondents with a deeper sense of self-awareness, appreciation, and global interconnectedness:

". . .I think I'm quite an introvert, but actually I do enjoy spending time with a select few people, so I think I understand myself a little bit better, and I think. . . I don't know whether I'm more laid back or whether I appreciate that I'm quite laid back in terms of. . . although I had worries and I think everybody had worries, that I was able to still function and not disregard them but certainly continue despite any worries or anxieties around it, so yeah." (Joely, T2)

"I feel like I'm more aware of what I do impacts other people, because you know before, you kind of just go about your day and you don't think if I touch this thing that might kill somebody [laughing]. Like I feel a lot more connected to the rest of the world now, about how my actions impact other people." (Chloe, T2)

**Sub-theme two:** Being kind and keeping structure
Being kind to oneself and protecting time to engage in self-care activities was important for maintaining the emotional wellbeing of T2 respondents:

". . .I was making sure that every day I went out for a walk as well which was very helpful. Luckily, I live somewhere vaguely rural so at the start I was trying to maintain kind of. . . some kind of normality." (Lillian, T2)

Being required to spend more time at home, due to social distancing restrictions, was opportune for T2 respondents to engage in rewarding activities which would not have otherwise been prioritised pre-pandemic:

"A lot of gardening, yeah we planted a lot of vegetables in the garden and it's kind of overtaken the garden, but yeah we did all of that which we didn't have time. . . like last year we didn't have time to do, but obviously we were out, spend days in the garden so we've managed to get loads of stuff done in the garden and get stuff done around the house, which is good." (Crissey, T2)

For these women, engaging in Do-It-Yourself activities allowed respondents to assert control amidst the uncertainties of the pandemic:

". . .just like taking control over the situation as much as you can, realising that there are certain things that I can't control, so there's no point in worrying about them or like spending lots of my energy thinking about them, and trying to focus on things that I can control, and having little things to look forward to as well. . . so things like a delivery coming you know that's got something in for the baby and I'd be thinking "oh right, that's coming on this date" or whatever. You know, just trying to look at little things that like "oh we're doing this family quiz" or whatever, that's something to look forward to." (Xenia, T2)

Focusing on preparations for baby's arrival allowed mothers to distract from COVID-19 related challenges and to instead refocus on activities and projects which were meaningful and gratifying:

". . .buying books to read up about what happens during pregnancy and what to do when you've got a baby and stuff like that. I suppose the kind of to keep me busy, we focused on something new." (Janet, T2)

"We just did things in preparation for the baby coming, like I did the nursery, ordered furniture and stuff and put the furniture up and all that kind of stuff, just getting things ready for the baby arriving and also things around the house that we haven't gotten around to doing, so gave me more time to do that." (Zanthia, T2)

Finding new ways to build a routine enhanced feelings of acceptance considering COVID-19 uncertainties:

"I like to have a bit of structure, have a plan, not for every single day, but. . . It's a lot of time to fill when you're on your own and you're not getting any adult interaction. Like when you're at work, you see adults every day and I think when I'm on maternity leave, I like to see adults or a coffee or at a class or a group or something. But actually, not knowing whether that would happen is quite scary because I can't bear the thought of spending a year with just a baby. That sounds horrible, 'cause I love my babies, but I do need some adult interaction as well." (Daisy, T2)

**Theme 2**: Robbed of our pregnancy

Respondents expressed deep disappointment and sadness in being unable to engage in their pregnancy identity through shared rituals with loved ones, positive interactions with the public, and public recognition of their identity as a mother-in-waiting. For some mothers, the pandemic had devastated their last experience of pregnancy. The majority of T2 women felt

unaffected by the easing of social distancing restrictions, which left these respondents feeling excluded. Other difficulties identified during interviews included balancing home and working life commitments and intense anxieties over the potential of contracting the virus. Individual differences into national restriction adherence left T2 women feeling morally conflicted and at times, exposed to relational confrontation.

**Sub-theme 1:** Unmet expectations and feeling guilty

Women at T2 expressed disappointment in having had their experience of pregnancy over-shadowed by the uncertainties of the COVID-19 pandemic:

"I have a bit of sadness because I missed out on quite a lot of things that pregnant women would have enjoyed and benefited from in a coronavirus-free world, like free swimming and things like going on the bus and people offering you a seat, it's all really pathetic little things like that, but just missing out on that experience, the full experience. . ." (Janet, T2)

For T2 respondents, lack of visibility during pregnancy was believed to contribute towards feelings of disappointment, with the pregnancy experience being very closely intertwined with group responses and connection:

"With my first pregnancy, everybody was asking about it and I could get excited with work colleagues and friends and people, but this time it was very, it was quiet. . . kind of just gets forgotten because you're not visible. So, it felt quite yeah, just the sort of hard slog rather than exciting." (Daisy, T2)

COVID-19 related uncertainties were perceived to dampen respondent's ability to enjoy their pregnancy:

"I felt like I've just chosen the worst time to fall pregnant [laughing] and it was a bit cursed like we finally managed to get pregnant again and then it was just the worst time possible, so I felt a bit sad that it was. . . and because I don't think I'll have another baby, it was kind of like this is the last time I'll be pregnant, and it is a disaster so. . . I feel a bit sorry for the baby that I have whinged the whole time and been stressed [laughing] and I haven't really enjoyed it" (Chloe, T2)

Feeling unable to partake in post-lockdown activities, too, was saddening for respondents:

". . .you did get a bit more fed up because you've seen people kinda getting back to some normality and going back to work, going back and doing things and you are still not able to do that. Maybe it would have been nice to get back to work, it would have been nice to see my colleagues before going off on maternity." (Audrey, T2)

For some women, regret was expressed regarding the timing of their pregnancy which was perceived to have been negatively affected by COVID-19 related stressors:

". . .maybe we should have delayed this [pregnancy] this was a bad idea, but then nobody knew this was going to happen. Um so yeah, I guess I think I had some negative feelings towards it that I wouldn't have if COVID haven't had happened." (Lillian, T2)

Difficulties balancing home and working life responsibilities was guilt-inducing, as parents struggled to balance conflicting priorities in the absence of formal childcare support:

"Trying to balance work and childcare, because he [oldest child] is still in nursery so it is not like you could leave him at the table to do some work, he wanted to play with me otherwise, you would be putting him in front of the TV which you felt guilty about... so yeah, it was definitely much more difficult for the work-balance relationship really." (Audrey, T2)

Awareness of being considered clinically vulnerable plagued T2 respondents with feelings of guilt for leaving the safety of their home, and crippling anxiety for the potential of having contracted the disease:

"I went into the town and then I felt so guilty, apologising to all the staff for even being there. That night, when I got home, I found out that I was in the highest [coronavirus risk] bracket; I shouldn't have even done any of that [leaving the house]. Like honestly, my heart was pounding, I had palpitations and I felt so short of breath, I thought I had coronavirus." (Selina, T2)

**Sub-theme 2:** The voice of others feels loud
For some T2 respondents, toxic positivity worsened feelings of being invalidated:

"...my best friends who are not currently pregnant, they are sort of unbearably and relentlessly positive... you're not able to complain or you're not allowed to have a low mood if you know what I mean, depending on who I was talking to I would either feel dismissed or supported" (Selina, T2)

Individual differences in social distancing adherence caused moral, emotional, and relational conflict for respondents who were confused and unsure how to navigate easing restrictions:

"...it's been pretty confusing and the information on what pregnant people can do is pretty vague. I'm being told different things by different people about whether we can do anything or what we should be doing, so I found it quite stressful and especially seeing everybody else going out and people expecting us to get back into socialising and I don't really feel comfortable doing that, so it's... it's been quite tricky and I've had to have a few awkward conversations with people just to say I'm... I'm not ready to get back out there just yet." (Chloe, T2)

On the other hand, feeling respected for the precautions one took served to build rapport and respect:

"My employers have been really understanding to be honest, because of I did receive an email from the deputy head saying "technically you can come into work now", but you know "we'll understand that you come in on public transport and stuff like that, so if you'd rather work from home then that's okay" so that was good of them really..." (Xenia, T2)

Leaving the house was occasionally a point of contention for members of the public:

"I did have a man tell me I shouldn't be out on a crossing, I was crossing the road and he told me I shouldn't be out in my condition on my way to work [laughing], and I said "I'm not actually ill, I'm just pregnant" [laughing] and he just kind of looked at me and I walked off." (Crissey, T2)

The opinions of others were also expressed in force online:

"...[I saw] a lot of posts [online] that were quite judgemental about pregnant women saying like "you shouldn't go to the shops" and things like that, "you should be shielding" which obviously isn't true." (Daisy, T2)

Regarding the easing of social distancing restrictions at T2, mixed feelings were expressed by respondents which varied between gratitude for renewed feelings of freedom and connection with others, and Anxieties and uncertainties regarding future lockdown restrictions and virus contraction:

"As soon as restrictions were officially relaxed, I was out seeing friends in open areas..." (Joely, T2)

"...lockdown easing didn't really mean anything to me. Do you know what I mean, that I wasn't really able to take advantage of any of it, anyway, and it would have been too high risk to do so." (Selina, T2)

**Sub-theme 3:** Left in the dark
News reports were a source of anxiety and distress for T2 respondents:

"I was watching the news constantly and then I was crying, that's when I was anxious and worried" (Emma, T2)

Uncertainties related to the COVID-19 pandemic were also conducive to feelings of depression and anxiety:

"I hate "hoping for the best" because I just want to be sure. Not knowing what to expect going to hospital, I think you already don't know what to expect as a first-time mum, but it was just heightened with lockdown stuff going on and being on your own, going for induction on your own is just a depressing scenario." (Janet, T2)

"I think like heightened the anxiety at the beginning... maybe a little bit more anxiety when it was eased because of the uncertainty of what's happening around us." (Francesca, T2)

Feeling forgotten and untended was a complaint raised consistently at T2. These respondents reported a desire for a dedicated service to support perinatal women during unprecedented times, for one's emotional wellbeing and for lay summaries of how changing guidance impacts specific Trusts and pregnant women:

"...[there has been] a more pastoral oversight of all the women who would have been affected by that change, especially as we went into lockdown and everything became more uncertain and women were more anxious and everything became more unknown, I think that there could have been a sort of check-in service just to keep people updated with the changes as they took place." (Selina, T2)

"I wasn't given any information... I had to make my own enquiries, which I found that quite difficult because I feel like for a new first-time mum, I think you don't know what to ask so it would be good if they just pointed you in the right direction, said "oh you know this is available and that's available", but I haven't had that." (Zanthia, T2)

Inconsistencies in guidance led to disillusionment with and distrust towards the Government:

"I don't really trust what's being said if it's from the government. . . It's hard to know what's safe to do and what isn't, I don't really know what those figures are anymore 'cause they're not giving out the daily figures. I don't really feel like I have a grasp on how the pandemic is going in our country anymore." (Chloe, T2)

"I think it's because everything changes so quickly and as soon as you read one thing, there seems to be another thing come up, so you can never fully trust what something says because they could just change it. So, yeah there's just not a huge amount of trust that what you're actually reading is true. . ." (Crissey, T2)

## 4. Discussion

The current study used a recurrent, cross-sectional, thematic analysis to explore pregnant women's psychological experiences during the COVID-19 pandemic. Pregnant women at both timepoints reported having more time to engage presently with their lives and pregnancy, in the absence of busy work and life schedules which were perceived to be oppressing pre-pandemic. Pre-pandemic literature links mindfulness practice with positive mental health outcomes [53]. A study conducted with the public during social distancing restrictions found a significant relationship between mindfulness-based practices and both physical and mental health [54]. Whereas negative interpretations of the pandemic and engaging less in daily mindfulness practice was linked with deteriorated mental health [55]. Mindfulness is a cost-effective therapeutic technique which has shown utility in clinical and community settings [56]. Due to the pre-pandemic effectiveness of mindfulness in protecting emotional wellbeing, current findings suggest integrating mindfulness-based support groups and resources in community settings and online as signposting resources may offer antenatal mental health benefits.

Irrespective of timepoint, most perinatal women felt their pregnancy was overshadowed by the COVID-19 pandemic. At T1, frustrations and grief were expressed in being unable to engage in social pregnancy rituals, which would have otherwise cemented one's maternal identity. These findings were reflective of antenatal interviews conducted during the COVID-19 pandemic in other high-income settings e.g., Australia [57]. Similar feelings of grief were also expressed at T2, though focus shifted to one's inability to interact with strangers about one's pregnancy, which again invalidated their developing maternal identity. Offering: i) an extended period of maternity leave for women who were pregnant and who gave birth during national lockdown restrictions; and/or ii) campaigning to encourage women to engage in perinatal rituals post-pandemic, may offer therapeutic benefits to those adversely affected by the pandemic.

Most respondents agreed extended periods of rest, caused by social distancing restrictions, enabled respondents to bond more deeply with their foetus. Relatedly, bonding was facilitated by the mother being better able to listen to her physical and emotional needs, in the absence of pre-pandemic distractions. Higher prenatal bonding significantly predicts bettered infant socio-emotional development; improved attachment quality; lower parent-reported colic rating; and more positively perceived infant temperament at twelve months postpartum [58]. Pre-pandemic, length of maternity leave was positively correlated with self-reported ratings of overall health, and of maternal-infant bonding [59]. Postnatal women interviewed during the early stages of the COVID-19 pandemic expressed a desire for an extended period of maternity

leave so that those affected by the pandemic could engage in important perinatal rituals [34]. In addition to offering mental health benefits to mothers negatively impacted by national lockdown restrictions, extending a period of maternity leave may also offer indirect, longstanding physical and mental health benefits to one's infant through improved opportunities for bonding.

Exercising and spending time outside were particularly important for maintaining mental wellbeing in the current study. T2 respondents, especially, felt lockdown restrictions had resulted in them adopting healthier lifestyles than they would have achieved pre-pandemic. Exercise, and particularly outdoors, has been strongly linked with improved mood and reduced symptoms of anxiety, anger, and depression [60]. In the current study pregnant women were consistently able to acquire more active lifestyles during the pandemic, perhaps due to their reinstated recreational time.

Respondents experienced deteriorated mental health due to social distancing restrictions. T1 women felt imprisoned by lockdown restrictions, which was also identified in a similar study conducted with postpartum women [34]. This experience also substantiates rapid review literature which has found prolonged periods of quarantine to be associated with poorer psychological outcomes [61]. T1 respondents also took issue with provision of inadequate mental health services. Pregnancy and services for children aged 0–5 services were de-prioritised in favour of pandemic efforts [62], which has been thought to have resulted in negative indirect health costs e.g., via inability for health visiting services to identify children at serious risk of harm [63]. For T2 respondents, toxic positivity from friends left women feeling invalidated. Feeling negatively judged and insufficiently supported substantiates feelings of guilt and shame [64]. In a study of postnatal women, a desire for pro-active mental health campaigns to be targeted in support of perinatal populations during health crises was revealed [34], which was also salient among T2 respondents in the present study. A campaign which: i) validates women's emotional difficulties, ii) provides appropriate signposting to relevant support services, iii) provides practical guidance on how changing restrictions affect them, and iv) provides emotional guidance to support persons on how to approach mental health conversations, would serve to improve perinatal mental wellbeing during health crises and in para-pandemic society.

In response to easing social distancing restrictions, T2 respondents were apprehensive about future lockdowns and contracting COVID-19. Anxieties were well justified, as social distancing practices are widely used, highly successful preventative measures against novel virus contraction. Anxiety was exacerbated due to an awareness of being 'vulnerable' to the outbreak and having been advised to 'shield' during initial lockdown restrictions [65]. For these women, perceived insensitivity to one's precautionary measures against COVID-19 exposure caused relational strain. In contrast, feeling respected in one's decision manifested trust and rapport. For T2 only, this was associated with an increased awareness of one's global connectedness. In other published literature, a greater sense of personal responsibility for one's impact on others, and a greater 'group mentality' predict more stringent compliance with social distancing restrictions [66]. Current findings suggest promotional efforts to adhere with social distancing restrictions might benefit from focusing on emphasising compassion and consideration for one another's boundaries, in addition to reaffirming one's group social identity and responsibility [67].

Some T1 respondents felt without purpose as social distancing restrictions left them derelict of adult interaction. These findings were mirrored among postpartum women: who were exposed to chronic parenting stressors with regards to home schooling and childcare commitments for multiple children [34]. Prioritising formal education is vital for narrowing intellectual and social developmental gaps between children of different socioeconomic backgrounds.

The current study reinstates the importance of prioritising formal childcare services to optimise parental mental wellbeing [34] as well as early child development. Other mothers found solace in re-establishing structure and control through routine setting. Regaining control manifested as i) learning new skills (T1), completing home-improvement projects (T2), immersing in *'nesting'* behaviour (T1, T2), and disengaging from melancholic pandemic reporting through media platforms (T2). Maternal accounts in this study reflect resilient and less resilient efforts to manage COVID-19 uncertainties. Active coping strategies e.g., learning new skills, are important for maintaining emotional wellbeing and *'bouncing back'* from times of adversity [68]. Mental health campaigns should be designed and implemented which draw on elements of resilience and active coping with an aim to build on principles of *'what works'* when promoting self-care in relation to one's psychological wellbeing [69].

T2 respondents felt disillusioned and untrusting of the government due to frequent *'knee-jerk'* reactions to COVID-19 spread and mortality. Blanket guidance left pregnant women feeling abandoned and neglected: again, reflective of postnatal accounts [34]. The UK governmental response to the pandemic has been argued to be among the worst, globally, with one of the most severe death rates per 100,000 population [70]. Changeable social distancing guidelines were perceived to be confusing and neglectful of maternal needs [34]. Although stringent social distancing restrictions are necessary to protect public health [71], the current study suggests that greater priority should be placed on open and transparent dissemination of necessary guidance. Specifically, implemented guidelines should be clearly communicated and justified, whilst acknowledging uncertainty to maintain public relations [70]. Secondly, recognising and addressing the needs of more vulnerable populations e.g., perinatal women, is essential for maintaining respect, trust, and emotional wellbeing.

T2 respondents felt negatively judged and stigmatised by the public for being perceived to be placing their unborn baby at risk of contracting COVID-19. Receiving negative judgement from others may be a manifestation of *'intensive motherhood'* whereby an individual is perceived to be a *'bad mother'* for not engaging in risk-averse behaviour [72]. Justifications for stigma can be fear-related (i.e., disease contraction), belief-related (supra-natural/religious), and blame-related (to self or others) [73]. Negative public perceptions of mothers being in public during the pandemic may have been fuelled by initial social distancing guidelines which invoked pregnant women to *'shield'* [18]. Despite being withdrawn, this guidance had withstanding impacts on pregnant women's concerns about interacting as national lockdown restrictions eased. Feeling morally conflicted may have contributed to feelings of guilt in maternal accounts in the current study. In other domains of parenthood e.g., infant feeding, moral conflict has been a notable prerequisite for these emotional experiences [64]. Shifting from person-centred to collective-focused efforts to protect one another and prevent COVID-19 spread may serve to improve overall compliance with guidelines [66], dissipate moral conflict, and consequential negative affect.

## 4.1 Strengths, limitations, and future directions

Using a recurrent, cross-sectional, thematic analysis allowed for nuanced psychological experiences within the context of changing policy to be captured. Similarities were identified across timepoints, with increased feelings of frustration and ill-justice reported as initial social distancing restrictions were eased. Current findings add to the substantial body of literature demonstrating that untailored social distancing legislation has had dire consequences for the mental health of pregnant women. Although social distancing and 'lockdown' measures have been found in certain circumstances to provide time and space for bonding between parents and newborns [74], the findings from this study offer additional considerations for policy and

practice so to better protect perinatal mental health during times of unprecedented national strain. Telephone and virtual (e.g., Zoom) interviewing allowed for nationwide participation. Broadly reaching recruitment enabled the research team to represent the voices of women from diverse counties, which improved transferability of conclusions drawn.

This study was conducted in the very beginning of the pandemic situation in the United Kingdom, and therefore has merit in capturing those initial experiences of the pandemic when so much was yet unknown. It also took place before vaccination became available to all persons and before vaccination against COVID-19 was endorsed for pregnant and breastfeeding women. We know from other analyses conducted on national surveillance data in England [75] that ethnicity, age, and relative deprivation affected COVID-19 vaccination uptake amongst women of reproductive age. Further explorations of experiences of psychological health and wellbeing are required to understand how temporal, experiential changes may have occurred as pandemic restrictions changed as well as how availability of vaccination may have mediated experiences.

Attrition and reasons for attrition, too, were not routinely recorded for the current sample. Despite initial enthusiasm for participation, recruitment for both timepoints proved onerous, and with the relative unpredictability of birth, some women had to be included into their early postpartum and retrospectively reflect on their antenatal experiences, which could have introduced bias into the results. Nuances exist between these populations with regards to the effects of social distancing restrictions e.g., prohibition of partners from maternity suites versus pregnancy scans [34, 74], pregnant women initially being advised to shield due to their perceived vulnerability [18]. Our sample was also–on the whole–highly educated, something noted in much on-line research, and may affect generalisability. Finally, recruitment for the current project spanned from the 13th July– 2nd September 2020, meaning that retrospective accounts of the transitional experiences of pregnant women were captured inconsistently against national social distancing guidelines implemented (see Table 1 for summary of UK national lockdown restriction implementation by date). Heterogeneity in emplaced restrictions at a health service, county, and national level during this time may have unintentionally confounded interview data through nuances in social freedoms at time of data collection.

## 5. Conclusion

The current study used recurrent cross-sectional thematic analysis to explore psychological experiences of pregnancy during the COVID-19 pandemic. Irrespective of timepoint allocation, women who engaged in mindfulness-based activities appeared to benefit from engaging in self-care and eliciting control over their lives. Promotion of active coping strategies and mindfulness techniques in community settings and online are thus recommended. Pregnant respondents felt stripped of their identity as a new mother due to national lockdown restrictions. An active campaign to encourage perinatal rituals post-pandemic and extending periods of maternity leave for those adversely affected by the pandemic are recommended to support maternal emotional wellbeing and infant development outcomes. The majority of respondents reported a deterioration in mental health due to COVID-19 restrictions. This could have of course been due to the pandemic circumstances, or it could have been due to the pressures of new motherhood. Whilst further exploration is required to disentangle these two concepts, the likelihood is that a combination of the two contributed to these negative experiences. Implementing a national campaign which acknowledges the emotional difficulties experienced by pregnant women, which provides relevant signposting to mental health services and practical guidance about perinatal-specific social distancing restrictions would serve to improve perinatal satisfaction with healthcare support and better emotional wellbeing. Group-identity,

opposed to individual-level responsibility in campaigning to reduce the spread of COVID-19 may prove more effective, according to findings from the current study. The current study highlights the need for transparent and consistent communication of changing social distancing guidelines and appropriate tailoring of guidelines to more vulnerable populations, such as pregnant mothers.

## Acknowledgments

We would like to acknowledge and extend thanks to all respondents who took the time to participate in interviews during this difficult time. We would also like to thank Miss. Hannah Bass and Miss. Phoebe Civil (both formerly of the University of Liverpool) for their early contributions to data analysis.

## Author Contributions

**Conceptualization:** Leonardo L. D. De Pascalis, Joanne A. Harrold, Victoria Fallon, Sergio A. Silverio.

**Data curation:** Leanne Jackson.

**Formal analysis:** Leanne Jackson, Anastasija Podkujko, Monic Gaspar.

**Investigation:** Leanne Jackson, Sergio A. Silverio.

**Methodology:** Leanne Jackson, Siân M. Davies, Anastasija Podkujko, Monic Gaspar, Sergio A. Silverio.

**Project administration:** Victoria Fallon, Sergio A. Silverio.

**Resources:** Joanne A. Harrold, Victoria Fallon, Sergio A. Silverio.

**Software:** Leanne Jackson, Laura K. Soulsby.

**Supervision:** Victoria Fallon, Laura K. Soulsby, Sergio A. Silverio.

**Validation:** Victoria Fallon, Sergio A. Silverio.

**Visualization:** Sergio A. Silverio.

**Writing – original draft:** Leanne Jackson, Laura K. Soulsby, Sergio A. Silverio.

**Writing – review & editing:** Siân M. Davies, Anastasija Podkujko, Monic Gaspar, Leonardo L. D. De Pascalis, Joanne A. Harrold, Victoria Fallon.

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
