## [Decision Letter · Decision Letter 0]

21 Feb 2023

PONE-D-23-01119The antenatal psychological experiences of women during two phases of the COVID-19 pandemic: A recurrent, cross-sectional, thematic analysis.PLOS ONE

Dear Dr. Silverio,

Thank you for submitting your manuscript to PLOS ONE. After careful consideration, we feel that it has merit but does not fully meet PLOS ONE’s publication criteria as it currently stands. Therefore, we invite you to submit a revised version of the manuscript that addresses the points raised during the review process. Please address the reviewer's comments as best as possible, as I will strongly consider them in my decision. I noted in my review that the paper only interviewed women in the third trimester and wanted to know why a broader sample across pregnancy stages wasn't considered. Pregnancy experiences vary between pregnancy stages and also between first and second time mothers-the latter I felt needs to be addressed in the paper.

The chosen sample of respondents must also be carefully interpreted, given the selective sampling frame which influences the respondent's insights. An added perspective could be identifying how ethnicity shaped the respondent's experience, which would be insightful to potential readers.

Table 2 is useful to get a sense of broad UK restrictions; however, it's left to the reader to interpret the degree of disruption subjectively these may have caused. Considering our international readership, please contextualise the health system and social disruption pertinent to the study population.

In conclusion, I would also encourage the authors to reflect that the impact of the pandemic lockdown was seen differently in 2020 compared to later years when vaccination was available as an intervention. Given the evolution of the pandemic over three years(2020-2023) and our understanding, you may like to provide some lines reflecting on these changes and how the study results are retrospectively viewed.<o:p></o:p>

We look forward to receiving your revised manuscript.

Kind regards,

Danish Ahmad, MBBS,MSc,MNAMS,PhD

Academic Editor

PLOS ONE

Journal Requirements:

4. Please amend your manuscript to include your abstract after the title page.

**Additional Editor Comments:**

Dear Sergio,

As the handling editor for your paper,I was pleased to review the paper and the reviewers report. I contacted close to 16 reviewers and only secured one who provided a prompt reviewer report. I would be rendering an enditorial decision basis my review of the paper and the reviewer's and will aim for a quick turn around.

Best

Danish

Reviewers' comments:

Reviewer's Responses to Questions

**Comments to the Author**

1. Is the manuscript technically sound, and do the data support the conclusions?

Reviewer #1: Yes

2. Has the statistical analysis been performed appropriately and rigorously? 

Reviewer #1: N/A

3. Have the authors made all data underlying the findings in their manuscript fully available?

Reviewer #1: No

4. Is the manuscript presented in an intelligible fashion and written in standard English?

Reviewer #1: Yes

5. Review Comments to the Author

Reviewer #1: Thank you for the opportunity to review this interesting paper.

Introduction

Why would you compare pregnant women to nulliparous adults? Would you not compare them to non-pregnant people?

1.2 I think you need a bit more of the way the disease was spreading prior to the pandemic declaration, currently the text goes form the first case to pandemic. At the time the Public Health Emergency of International Concern was a big deal.

The WHO in China first noted reports of viral pneumonia in Wuhan on 31st December 2019. I think you should identify the rapid spread of the disease and that by 31st Jan 2020 the WHO declared a Public Health Emergency of International Concern (WHO Situation report no.11), by the time the pandemic was declared on 11th March (WHO satiation report no. 51) there were more than 118,000 cases and almost 4300 deaths in 23 countries.

1.3 could “Antenatal and 0-5 services…” be rephrased to read more clearly: Antenatal and services for children aged 0-5……

The phrase “, …was ineffective for building the necessary report to disclose…” probably should state rapport

Informal support (eg maternal grandmother)- could this not be broadened to grandparents? Might be more inclusive.

2. The paragraph about participation exhaustion making recruitment difficult and yet you state previously that recruitment was over-subscribed. This would appear to be contradictory. I am confused regarding the need for the additional recruitment strategy and that some women were recruited in September. Time point 1 was March 2020 and time point 2 was July 2020 (when restrictions lifted). As it was a one off interview what was the issue with attrition? Could you not have gone back to the random generator and recruited other women? How postnatal were the postnatal women? Obviously this is a huge source of potential recall bias, was this strategy really the most effective given the initial interest from 72 women (3 times the number actually interviewed)?

Table 1. Interesting that only 3 women did not have a degree at all and everyone at T2 had at least one! Overall 25% had Doctorates (6/24) although usually samples often contain those with highest educational attainment, this group would appear particularly so. Interesting when compared to the UKOSS evidence around women who got COVID.

Is there a typo in time point 1 Ceinwen ‘s quote?

Discussion: Many of the quotes and feelings could have been due to the pregnancy per se. Overall the findings are appropriately presented and discussed

Strengths and Limitations: The authors acknowledge this study adds to the ‘substantial body of literature around untailored social distancing legislation” and the ‘dire consequences’ for the mental health of pregnant women. But is this different from the catastrophic impact on mental health social distancing had for the general population, and the body of evidence around this? It could be argued the additional time to look after themselves and bond with their unborn child might counterbalance the impact of social isolation.

I still think further explanation is required around attrition given the positive spin around initial over recruitment. You have not discussed the impact of potential bias associated with recruiting in this way or dealing with the attrition. Interestingly the lack of “baby showers” and other social rituals around pregnancy indicate how important these are but that ‘baby showers’ are a relatively recent invention and yet clearly are the ‘norm’ for women in pregnancy now.

When you conclude that most respondents experienced deterioration in mental health during the pandemic, could this be due to the pregnancy, or be due to the same influences as the general population?

6. PLOS authors have the option to publish the peer review history of their article (what does this mean?). If published, this will include your full peer review and any attached files.

Reviewer #1: **Yes: **Annette Briley

---

## [Author Response · Author response to Decision Letter 0]

25 Mar 2023

We have provided a separate document with a point-by-point response.

---

## [Editor Report · Decision Letter 1]

17 Apr 2023

PONE-D-23-01119R1

The antenatal psychological experiences of women during two phases of the COVID-19 pandemic: A recurrent, cross-sectional, thematic analysis.

PLOS ONE

Dear Dr. Sergio A.Silverio 

Thank you for submitting your manuscript to PLOS ONE. After careful consideration, we feel that it has merit but does not fully meet PLOS ONE’s publication criteria as it currently stands. Therefore, we invite you to submit a revised version of the manuscript that addresses the points raised during the review process.

I thank the authors for working on the editor and reviewer comments and providing a revised manuscript. I can recommend it for publication barring one additional change I encourage the authors to make in another revised submission which would receive an expedited review. Reading the results section and themes linked to the period provides key insights. While the authors flag the timepoint, it is slightly confusing in the final revised paper to know which theme/subtheme was linked to the particular period, especially for T2 themes number two onwards. Moreover, the results haven't currently provided both time point 1 and 2 themes together, allowing for side-by-side comparison. While the discussion elaborates on themes, visually comparing both themes allows for greater comprehension of the results and enhances linkages in the results section

I suggest the authors add a table providing T1 and T2 themes together in the results section, preferably before the T1 theme's results start. For the author's reference, I have attached an example of such a  table. Please consider providing  a few additional lines in the results section to start introducing the table and providing readers with information on how the results section is presented viz that the results first provides time period themes one followed by period two themes etc

In future, tracking revisions in the paper against reviewer comments would be improved by providing line and page numbers where changes are made.

We look forward to receiving your revised manuscript.

Kind regards,

Danish Ahmad, MBBS,MSc,MNAMS,PhD

Academic Editor

PLOS ONE

Journal Requirements:

Additional Editor Comments:

I thank the authors for working on the editor and reviewer comments and providing a revised manuscript. I can recommend it for publication barring one additional change I encourage the authors to make in another revised submission which would receive an expedited review. Reading the results section and themes linked to the period provides key insights. While the authors flag the timepoint, it is slightly confusing in the final revised paper to know which theme/subtheme was linked to the particular period, especially for T2 themes number two onwards. Moreover, the results haven't currently provided both time point 1 and 2 themes together or in one place, allowing for side-by-side comparison. While the discussion elaborates on themes, visually comparing both period themes allows for greater comprehension of the results and enhances linkages in the results section

I suggest the authors add a table providing T1 and T2 themes together in the results section, preferably before the T1 theme's results start. For the author's reference, I have attached an example of such a  table. Please consider providing  a few additional lines in the results section to start introducing the table and providing readers with information on how the results section is presented viz that the results first provides time period themes one followed by period two themes etc

In future, tracking revisions in the paper against reviewer comments would be improved by providing line and page numbers where changes are made
---

## [Author Response · Author response to Decision Letter 1]

17 Apr 2023

We thank the Editor for this suggestion. We have made the timepoints clearer in both the abstract and the methods section, and – as requested – have added a table of themes as per the timepoints into the results section.

---

## [Editor Report · Decision Letter 2]

19 Apr 2023

The antenatal psychological experiences of women during two phases of the COVID-19 pandemic: A recurrent, cross-sectional, thematic analysis.

PONE-D-23-01119R2

Dear Dr. Sergio A. Silverio

We’re pleased to inform you that your manuscript has been judged scientifically suitable for publication and will be formally accepted for publication once it meets all outstanding technical requirements.

Kind regards,

Danish Ahmad, MBBS,MSc,MNAMS,PhD

Academic Editor

PLOS ONE

Additional Editor Comments (optional):

I am pleased to accept the paper for publication and thank the authors for working with the journal through the review process.

The copy editing process should provide authors a chance to correct a minor typo in the abstract in the line "... after the initial lockdown restriction**st**"
---

## [Editor Report · Acceptance letter]

16 May 2023

PONE-D-23-01119R2 

The antenatal psychological experiences of women during two phases of the COVID-19 pandemic: A recurrent, cross-sectional, thematic analysis. 

Dear Dr. Silverio:

I'm pleased to inform you that your manuscript has been deemed suitable for publication in PLOS ONE. Congratulations! Your manuscript is now with our production department. 

Kind regards, 

on behalf of

Dr. Danish Ahmad 

Academic Editor

PLOS ONE